# Online Decision Based Visual Tracking via Reinforcement Learning

**Ke Song**   **Wei Zhang**[*]  **Ran Song**   **Yibin Li**
School of Control Science and Engineering, Shandong University, Jinan, China
`songke_vsislab@mail.sdu.edu.cn`
`{davidzhang,ransong,liyb}@sdu.edu.cn`

## Abstract

A deep visual tracker is typically based on either object detection or template matching while each of them is only suitable for a particular group of scenes. It is straightforward to consider fusing them together to pursue more reliable tracking. However, this is not wise as they follow different tracking principles. Unlike previous fusion-based methods, we propose a novel ensemble framework, named DTNet, with an online decision mechanism for visual tracking based on hierarchical reinforcement learning. The decision mechanism substantiates an intelligent switching strategy where the detection and the template trackers have to compete with each other to conduct tracking within different scenes that they are adept in. Besides, we present a novel detection tracker which avoids the common issue of incorrect proposal. Extensive results show that our DTNet achieves state-of-the-art tracking performance as well as a good balance between accuracy and efficiency. The project website is available at `https://vsislab.github.io/DTNet/`.

## 1   Introduction

As a fundamental task in computer vision, visual tracking aims to estimate the trajectory of a specified object in a sequence of images. Inspired by the success of deep learning in general computer vision tasks, recent visual tracking algorithms mostly used deep networks, particularly CNNs which extract deep representations for various scenes. Among these deep trackers are two dominant tracking schemes. The first one treats tracking as a detection task, which typically builds a deep network to distinguish the foreground target from the background [5, 25, 39]. The second one regards tracking as a template matching task and addresses it via a matching network such as Siamese network, which learns a general similarity function to obtain the image patch best matching the target [11, 15, 29].

The detection tracker continuously updates the network online with the image patch detected as the target by itself. The diverse appearances of the patches lead to a good adaptability of the tracker while the continuous update is inefficient for real-world tracking. Also, albeit occasionally, an incorrect detection in a frame which represents a noisy appearance of the target could mislead the tracker. The template tracker utilizes the initial appearance of the target as a fixed template to conduct the matching operation, which runs efficiently at the cost of adaptability.

Either the detection or the template tracker is merely suitable for a particular group of scenes. For instance, as shown in the top row of Fig. 1, due to the temporal occlusion within a frame, the detection tracker incorrectly captures the bicycle as the target in that frame and cannot recover from it in the succeeding frame. By contrast, the template tracker is robust to the temporal occlusion as it always looks back to the real target in the initial frame for delivering the matching. On the other hand, it

---

[*]Corresponding author

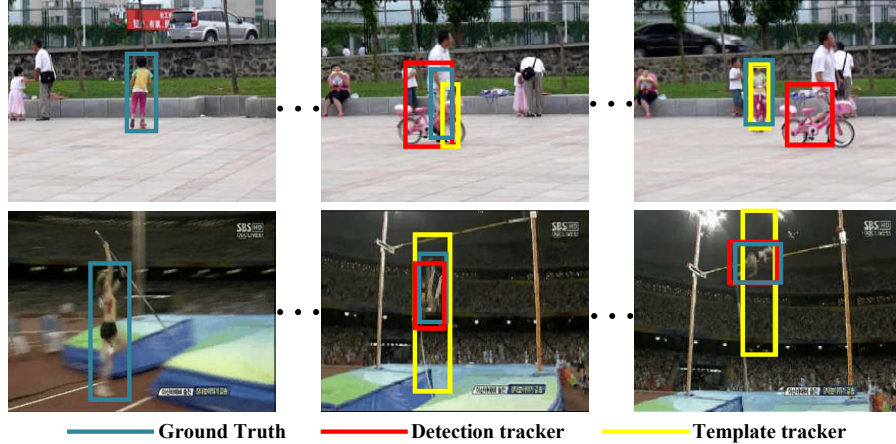

| ▬ Ground Truth | ▬ Detection tracker | ▬ Template tracker |

Figure 1: Failure cases of the detection and the template tracker caused by temporal occlusion and temporal deformation. The leftmost column shows the initial frames of the two videos.

is easy to understand that, as shown in the bottom row of Fig. 1, the template tracker is not reliable with the temporal deformation of the target while the detection tracker works well with it. Some recent works investigated various fusion schemes to pursue better performance [2, 17, 20, 34, 40]. However, directly fusing the two types of trackers together is not wise as they follow different tracking principles and thus cannot converge to each individual optimum simultaneously during training. Hence, it might be better to make them co-exist for handling different scenes alternatively.

Differing from previous fusion-based methods, this paper presents a framework of decision learning for the ensemble of the two types of trackers where we explore how to automatically and intelligently switch between them for tracking in different scenes. Specifically, our method makes the two trackers compete with each other through a hierarchical reinforcement learning (HRL) framework so that it can make a proper online decision to choose the tracker which captures the target better in the current scene. This idea is based on the common observation as shown in Fig. 1 that different types of trackers are merely good at tracking the targets in a particular group of frames.

We name the ensemble framework DTNet as it comprises a decision module and a tracker module as illustrated in Fig. 2. The decision module starts with a switch network that encodes the image patch inheriting from the previous frame and the target in the initial frame to decide whether the detection or the template tracker should be selected for the current frame. It is followed by a termination network which estimates the output of the tracker to generate a probability of terminating the current tracker. The switch and the termination networks in fact form a "Actor-Critic" structure [21]. Such intelligent switching between the two trackers repeats till all frames of the video are processed. We provide a specifically designed scheme for jointly training the decision and the tracker modules end-to-end via HRL.

Furthermore, to improve the detection tracker, a fully-convolutional classifier is learned to differentiate the target from the distracting content, Since it does not rely on a number of candidate proposals to predict the bounding boxes of the target, it actually avoids the issue of the incorrect prediction of such proposals that could mislead the tracker. The contributions of this paper are summarized as follows.

- We propose an ensemble framework which learns an online decision for visual tracking based on HRL where the detection and the template trackers compete with each other to substantiate a switching strategy.

- We develop a novel proposal-free detection tracker, which does not require the proposal of candidate bounding boxes of the target and thus make the discriminating course flexible.

- Our method demonstrates the state-of-the-art performance on several benchmarks. The ablation studies show that the decision mechanism composed of the switch and the termination networks can effectively select the proper trackers for different scenes.

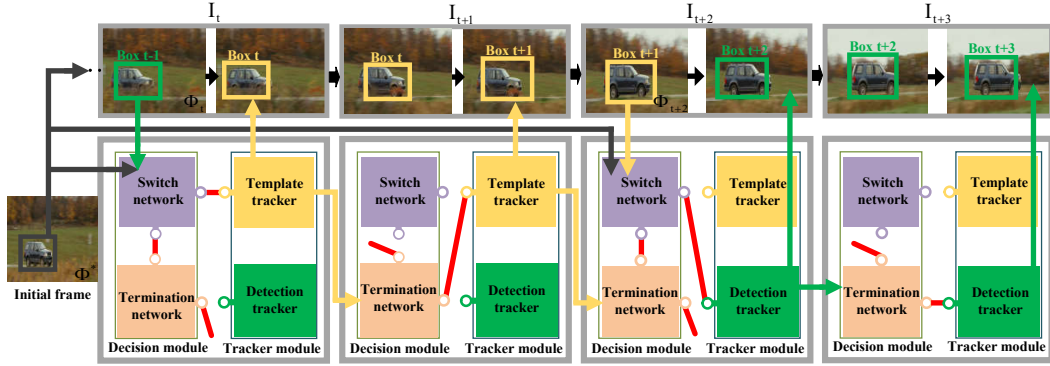

Figure 2: Overview of the DTNet

## 2 Related Work

**Detection trackers.** Trackers based on object detection in each video frame usually learn a classifier to pick up the positive candidate patches wrapping around previous observation. Nam and Han [25] proposed a lightweight CNN to learn generic feature representations by shared convolutional layers to detect the target object. Han $et$ $al.$ [14] selected a random subset of branches for model update to diversify learned target appearance models. Fan and Ling [13] took into account self-structural information to learn a discriminative appearance model. Song $et$ $al.$ [30] integrated adversarial learning into a tracking-by-detection framework to reduce overfitting on single frames. However, the occasional incorrect detection in a frame is still prone to contaminate and mislead the target appearance models.

**Template trackers.** Trackers based on template matching have recently gained popularity due to its efficiency, which learns a similarity function to match the target template with the image patch in the searching region of each frame. Tao $et$ $al.$ [31] utilized Siamese network in an offline manner to learn a matching function from a large set of sequences, and then used the fixed matching function to search for the target in a local region. Bertinetto $et$ $al.$ [4] introduced a fully convolutional Siamese network (SiamFC) for tracking by measuring the region-wise feature similarity between the target object and the candidate. Wang $et$ $al.$ [36] incorporated an attention mechanism into Siamese network to enhances its discriminative capacity and adaptability. However, these trackers are prone to drift when the target suffers the variations such as shape deformation and color change in appearance due to the fixed appearance of the template without an online update.

**Fusion-based trackers.** There exist some trackers adapting fusion strategies. The MEEM algorithm [40] proposed a multi-expert tracking framework with an entorpy regularized restoration scheme. And Li $et$ $al.$ [20] introduced a discrete graph optimization into the framework to handle the tracker drift problem. Wang $et$ $al.$ proposed MCCT [35] which selected the reliable outputs from multiple feature to refine the tracking results. Bertinetto $et$ $al.$ [3] combine two image patch representations that are sensitive to complementary factors to learn a model robust to colour changes and deformations. However, it is not easy to fuse multiple trackers significantly different in principle, as they can hardly converge to each individual optimum simultaneously during training. Unlike the fusion-based methods above, our method aims to learn an online strategy to decide which tracker should be used for each individual frame.

## 3 Method

As shown in Fig. 2, the proposed framework consists of two modules: the decision module and the tracker module. As the key component of the entire framework, the former contains the switch network and the termination network, which work together to alternatively select the template or the detection tracker which compete with each other in the tracking task and jointly form the tracker module. In the decision module, the switch network encodes an image patch $\Phi_t$ inheriting from the previous frame $I_{t-1}$ and the initial template $\Phi^*$, and then outputs a binary signal to select a tracker. A tracker can estimate the location of the target for the current frame $I_t$. The termination network

estimates the output of the tracker and generates a probability to decide if the framework should keep using the current tracker or terminate it, which makes the decision module avoid oscillating between the two trackers especially when they have similar accuracy. Note that if the termination network decides to terminate, it merely indicates that the current tracker in use does not work well while it does not necessarily means that the other tracker can performs better. Thus in this case, the switch network will still select a new tracker from the two candidate trackers instead of blindly switching to the other tracker currently not in use. Fig. 2 illustrates all of the 4 possible switching situations of the framework.

## 3.1 Decision Module

Given a set of states $S$ and actions $A$, the Markovian options $w \in \Omega$ consist of three components [1]: an intra-option policy $\pi : S \times A \rightarrow [0, 1]$, a termination condition $\beta : S^+ \rightarrow [0, 1]$, and an initiation set $I \subseteq S$. Here we assume that $\forall s \in S, \forall w \in \Omega : s \in I$ (i.e., the trackers are both available in all states). If an option $\omega$ is taken, then actions are selected according to $\pi_\omega$ until the option terminates stochastically according to $\beta_\omega$. For controlling the switch of tracker in an HRL manner, the decision module utilizes the termination policy together with the policy over options corresponding to the trackers. Let $Q_\Omega$ denote the switch network which can be viewed as a function subject to option $\Omega$ parameterized with its network weights $\theta$ and the termination network $\beta_{\Omega,\nu}$. A termination probability which decides if the current tracker in use should be terminated is estimated by $\beta_{\Omega,\nu}$ depending on option $\Omega$ and its network weights $\nu$. Specifically, we define $Q_\Omega$ as below to evaluate the value of option $\omega$ in a manner of hierarchical reinforcement learning:

$$Q_\Omega(s, w; \theta) = r(s, w) + \gamma U(s', \omega), \tag{1}$$

where $r(s, w)$ denotes the reward that the agent receives after implementing $\omega$ representing the option for selecting a particular tracker. $\gamma$ is the discount factor and $U(s', \omega)$ is the value of executing $\omega$ on a new state $s'$ related to the termination probability $\beta_{\omega,\nu}$, which is computed by combining the outputs of the switch and the termination networks:

$$U(s', \omega) = (1 - \beta_{\omega,\nu}(s'))Q_\Omega(s', w) + \beta_{\omega,\nu}(s')V_\Omega(s'), \tag{2}$$

where $\beta_{\omega,\nu}(s')$ is the termination probability on the state $s'$, and $V_\Omega$ is the optimal of the switch function which can be found by searching for the maximum of the switch function $Q_\Omega$ over option $\omega$:

$$V_\Omega = \max_\omega(Q_\Omega(s', \omega)). \tag{3}$$

If the current option, expressed as $\omega_{good}$, works well, the agent will not terminate it, which means that $\beta_{\omega,\nu}$ is close to 0. Thus based on Eq. (2), we have $U(s', \omega_{good}) \approx Q_\Omega(s', \omega_{good})$. If it is not a good option, according to Eq. (3), we have $V_\Omega = Q_\Omega(s', \omega_{good})$. In this case, the agent tends to terminate the current option, which means that $\beta_{\omega,\nu}$ is close to 1. Thus according to Eq. (2), we also have $U(s', \omega_{good}) \approx Q_\Omega(s', \omega_{good})$ as desired. Note that $U(s, \omega)$ is differentiable. Its gradient with respect to the weights $\nu$ of the termination network is expressed as:

$$\frac{\partial U(s', \omega)}{\partial \nu} = -\frac{\partial \beta_{\omega,\nu}(s')}{\partial \nu}(Q_\Omega(s', \omega) - V_\Omega(s')) + (1 - \beta_{\omega,\nu}(s'))\frac{\partial U(s'', \omega')}{\partial \nu}. \tag{4}$$

A similar form as in Eq. (4) can be derived by expanding $\frac{U(s'', \omega')}{\partial \nu}$ recursively. Here the state-option pairs $(s, \omega)$ in one time step is involved in the calculation. As shown in Fig. 2, the switch network $Q_\Omega(s, \omega)$ acting as the 'Critic' evaluates the value of options and provides the updating gradients for the network termination network $\beta_{\omega,\nu}$, which essentially acts as the 'Actor' and evaluates the performance of the tracker in use to decide if it should be terminated in the current frame so that the agent could optionally switch to the other tracker for the next frame. The weights $\theta$ of the switch network are learned the Bellman equation and the details will be given in Section 3.3.

## 3.2 Tracker Module

**Template tracker.** We adopt SiamFC [4] as the template tracker. The standard Siamese architecture takes as input an image pair containing an exemplar image $z$ and a candidate image $x$. The image $z$ represents the object of interest (e.g., an image patch centered on the target object in the first video frame), while $x$ is typically larger and represents the searching area in the subsequent video

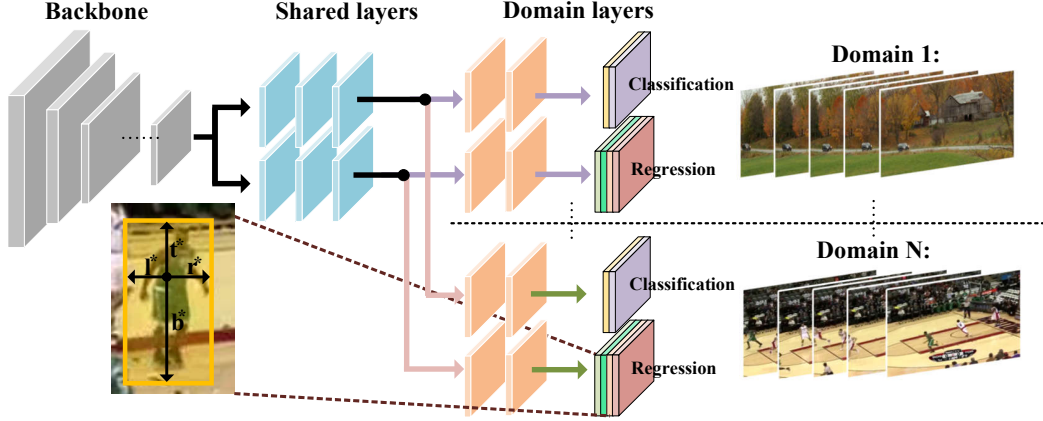

Figure 3: Architecture of the proposed FCT tracker

frames. The features of $z$ and $x$ are extracted by the same CNN $\varphi$ parametrized with $\tau$, which are cross-correlated as:

$$f_\tau(z, x) = \varphi_\tau(z) \star \varphi_\tau(x) + b \tag{5}$$

where $b$ denotes a bias term which takes the value $b \in R$ at every location, $\star$ represents the operation of convolution. Eq. (5) performs an exhaustive search for the pattern $z$ over the image $x$. The goal is to match the maximum value in the response map $f$ to the target location.

**Detection tracker.** To build a tracker based on object detection while avoiding the expensive process of proposal generation, we adopt a fully convolutional tracker, namely FCT, as shown in Fig. 3 which includes a classification branch and a regression branch. The classification branch predicts the location of the target and while the regression branch a 4D vector indicating the distances from the center of the target to the edges of its bounding box.

Given the feature map $F \in R^{H \times W \times C}$ of a backbone CNN and the sum $s$ of all strides applied in previous layers, each location $(x, y)$ in $F$ corresponds to $(\lfloor \frac{s}{2} \rfloor + xs, \lfloor \frac{s}{2} \rfloor + ys)$ in image. And we directly predict the class label and the regressed distances for each location in $F$ [32].

It is possible that the same class of objects are considered as targets in one sequence but background objects in another one. Due to such variations and inconsistencies, only using a typical classifier to simply assign "1" to the target and "0" to the background for all sequences is likely to cause conflicts across sequences [25]. Therefore, the proposed classification branch separates domain-independent information from the last domain-specific layer to capture shared representations via shared layers. Specifically, in each domain the location $(x, y)$ is considered as a positive sample if it falls into the groundtruth box and the class label $c^*$ is assigned 1. Otherwise, it is a negative sample (i.e. background) and the class label $c^*$ is set to 0.

The regression branch outputs a 4D vector $re^* = (l^*, t^*, r^*, b^*)$ where $l^*, t^*, r^*$ and $b^*$ denote the distances from the location of the target to the four edges of its bounding box as shown in Fig. 3. The tracker finally outputs the classification score map $c$ and the regression value $re$. The loss function for training is given as below:

$$L(c, r) = \frac{1}{N} \sum_{i=1}^{N} L_{cls}(c_i, c_i^*) + \frac{\lambda}{N} [Where_{\{c^*>0\}}] \sum_{i=1}^{N} L_{reg}(re_i, re_i^*), \tag{6}$$

where $N$ denotes the total number of the video frames for training.

### 3.3   Joint Training of Decision and Tracker Modules

In this section, we detailed the joint training procedure of the DTNet, in which the decision and the tracker modules are trained end-to-end. Given $K$ training sequences, for the $j$-th one we randomly extract a piece of training sequences $I_j = \{I_{1j}, I_{2j}, ..., I_{Tj}\}$ with the corresponding ground truth $G_j = \{G_{1j}, G_{2j}, ..., G_{Tj}\}$ in order, and each pair of adjacent frames is subject to a skip of $n(0 \leqslant n \leqslant 5)$ frames with some probability. The initial target is sampled around the ground truth

randomly in the first frame and regarded as the template. The switch network optionally evaluates the features encoded in the template and the observation inheriting from the previous frame and then selects a tracker. The reward during the switching process is defined as:

$$r_t(s, \omega) = \begin{cases} \eta_L \cdot D_{IoU}, \ IF \ (P_t > th_{hi} \ and \ P_t^* < th_{lo}) \\ \eta_L \cdot D_{IoU}, \ IF \ (P_t < th_{lo} \ and \ P_t^* > th_{hi}) \\ \eta_M \cdot D_{IoU}, \ IF \ (P_t > th_{hi} \ and \ P_t^* > th_{hi}) \\ \eta_S \cdot D_{IoU}, \ IF \ (P_t < th_{lo} \ and \ P_t^* < th_{lo}) \end{cases} \tag{7}$$

where $P_t$ is the intersection-over-union (IoU) between the predicted bounding box $B_t$ from the selected tracker and the ground truth $G_t$ and $P_t^*$ is the IoU corresponding to the unselected tracker. $D_{IoU}$ is the difference value between them. Actually, three cases are divided by the above setup of reward: (1) One succeeds while the other fails; (2) Both succeed; (3) Both fail. Accordingly, three enlarger coefficients are assigned in descending order, which leads to select the agent with higher accuracy while guides the tracking competition. The samples are collected by the unselected tracker respectively to update the corresponding network. In other words, we keep on training the worse one to maintain the competitive relationship between the two trackers. A new state $s'$ is updated for the current frame by the prediction. Then, the agent takes the probability of $\beta_{\omega,\nu}(s')$ to terminate the previous option and re-evaluate the value of options.

For the switch module, the 'Critic' model $Q_\Omega(s, \omega)$ can be learned using the Bellman equation [22], the learning process is achieved by minimizing the following loss:

$$L = \frac{1}{N} \sum_{i=1}^{N} (y_i - Q_\Omega(s_i, \omega_i; \theta))^2 \tag{8}$$

where $y_i = r(s_i, w_i) + \gamma(1 - \beta_{\omega_i,\nu}(s_i')Q_\Omega(s_i', \omega_i)) + \beta_{\omega_i,\nu}(s_i')V_\Omega(s_i')$. And the 'Actor' module $\beta_{\omega,\nu}$ updates as follows:

$$\nu = \nu - \alpha_\nu \frac{\partial \beta_{\omega,\nu}(s')}{\partial \nu}(Q_\Omega(s', \omega) - V_\Omega(s')). \tag{9}$$

Please refer to Algorithm 1 in the supplementary material available at the website mentioned in the abstract for the details of the whole training process.

## 4   Experimental Results

In this section, we conduct comparative evaluations on the benchmarks including OTB-2013 [37], OTB-50 [38], OTB-100 [38], LaSOT [12], TrackingNet [24], UAV123 [23] and VOT18 [18] with three considerations: 1) We compare the proposed DTNet with state-of-the-art trackers; 2) To demonstrate the effectiveness of the switch module, we compare the DTNet with some of its variants by employing different rackers; 3) We further compare our method with the trackers fused at the feature level to demonstrate the advantage of the decision-based strategy. Apart from the experimental results shown in this section, please refer to the website mentioned in the abstract for the supplementary results including the online visualization of the decision module of the proposed DTNet and the comparison with the state-of-the-art tracking methods.

**Implementation details.**  We build the switch and the termination networks by three convolutional layers and two fully connected layers, which receive the image patch of $84 \times 84$ as input. The sequences from VID [28] and Youtube_BB [27] datasets are used to train the DTNet including the decision and the tracker modules for $6 \times 10^5$ episodes with Adam optimizer. We set the capacity of the replay buffer to 5000, the learning rate to 0.0001, the discount factor $\gamma$ in Equ. 1 to 0.2, the batch size to 128 and $n_\kappa$ is set to $3 \times 10^5$. For the $\epsilon$-greedy algorithm, $\epsilon$ is set to 1 and decays to 0.1 gradually. The experiments were implemented in PyTorch on a computer with a 3.70GHz Intel Core i7-8700K CPU and two NVIDIA GTX 1080Ti GPUs. The average tracking speed is 36 FPS.

**Comparison with state-of-the-art trackers.**  We compare the DTNet (with FCT+SiamFC in this version) with the state-of-the-art trackers including CNN_SVM [16], SiamFC [4], DSST [9], ECO [8], SRDCF [10], SCT [6], HDT [26] and Staple [3]. In Fig. 4, we can observe that our DTNet achieves state-of-the-art performance in terms of the success rate and the precision on the OTB-2013, OTB-50 and OTB-100 datasets. It is noteworthy that although DTNet performs slightly worse than ECO, it

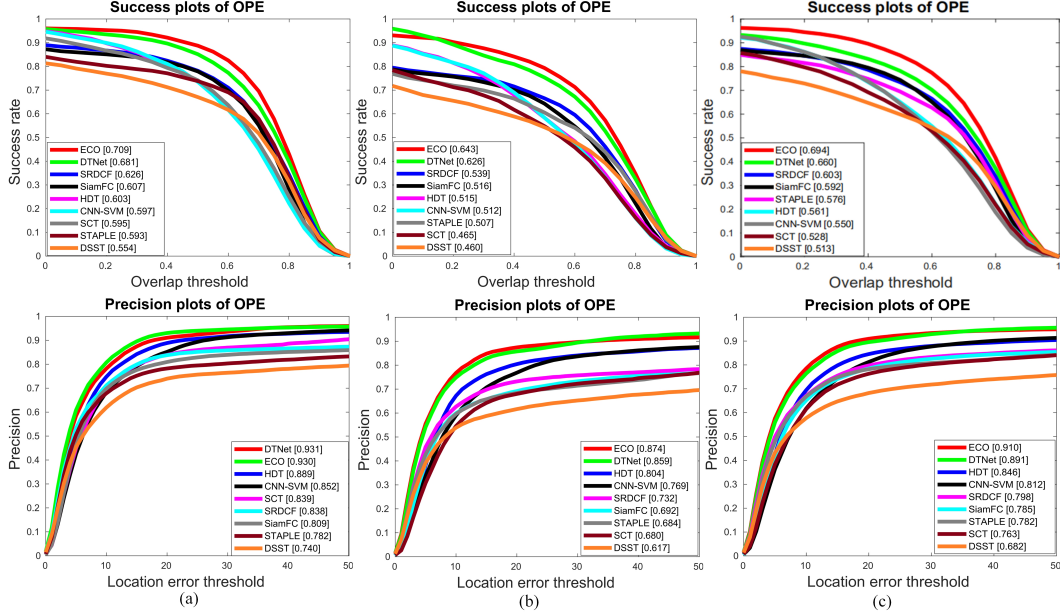

Figure 4: Results on OTB-2013, OTB-50 and OTB-100 in terms of success rate and precision

is much more efficient than ECO. The high performance of DTNet can be attributed to two aspects. First, the decision module intelligently selects a proper tracker for each frame instead of fusing two trackers that could conflict with each other. Second, we improve the original detection tracker by considering the domain knowledge, and makes the discriminating course more flexible through eliminating the candidate boxes.

**Comparison with variants.** We conduct ablation study to investigate the effectiveness of the DTNet. By comparing the quantitative results listed in the top half of Table 1 with those in its bottom half, we can see that the DTNet which combines two trackers always outperforms its ablated version which merely uses one tracker no matter what a single tracker is used.

To further validate the effectiveness of the decision module of the DTNet, We have included a manually designed rule based decision module for comparison. It is implemented by picking a particular tracker based on the confidence score of tracking subject to the thresholds set manually. The results are given in the eighth row of Table 1. Apparently, our automated decision module significantly outperforms such a handcrafted one which relies on handcrafted thresholds for tracker selection. Besides, our method is more efficient as it only performs each tracker once in the decision-making process while the handcrafted module has to carry out both trackers and use their output confidence scores for decision.

We also compare the DTNet with its variants by exploring different combinations of trackers including ACT [5], FCT, ATOM [7], SiamFC [4], CFNet [33] and SiamRPN++ [19]. It is noteworthy that ACT, FCT and ATOM are detection trackers while SiamFC, CFNet and SiamRPN++ are template trackers. We always combine a detection tracker and a template tracker to form a variant of the DTNet for comparison. The results show that the DTNet constantly outperforms each individual tracker in terms of AUC and precision on different benchmarks, which demonstrate the effectiveness of the decision module. Table 1 also clearly shows that the DTNet makes a good balance between the performance and the efficiency compared with its variants which have different combinations of trackers.

Furthermore, our framework can be easily extended to more trackers. For instance, the results of using three trackers including ACT, FCT and SiamFC are shown in the penultimate row of the Table 1. Considering both accuracy and efficiency, we use two trackers in the proposed DTNet.

**Visualization of the decision module.** Fig. 5 shows the visualization of the decision module during training where the outputs of the switch network, i.e. the $Q$ values for the SiamFC and the FCT trackers estimated via the HRL (see Eq. 1), are displayed on top of each frame. It can be seen that in the first frame, the template tracker SiamFC works well while the detection tracker FCT outperforms

Table 1: Comparison with the variants of DTNet on different benchmarks.

| Method | OTB2015 | | TrackingNet | | UAV-123 | | LaSOT | | VOT18 | | | Speed↑ |
|---|---|---|---|---|---|---|---|---|---|---|---|---|
| | AUC↑ | Prec.↑ | AUC↑ | Prec.↑ | AUC↑ | Prec.↑ | AUC↑ | Prec.↑ | A↑ | R↓ | EAO↑ | (fps) |
| ACT | 0.625 | 0.859 | 0.533 | 0.578 | 0.512 | 0.708 | 0.339 | 0.301 | 0.518 | 0.344 | 0.296 | 30 |
| FCT | 0.616 | 0.812 | 0.541 | 0.593 | 0.542 | 0.697 | 0.351 | 0.327 | 0.493 | 0.349 | 0.278 | 41 |
| SiamFC | 0.582 | 0.771 | 0.531 | 0.571 | 0.493 | 0.575 | 0.336 | 0.339 | 0.503 | 0.585 | 0.188 | **86** |
| CFNet | 0.589 | 0.777 | 0.507 | 0.529 | 0.468 | 0.538 | 0.275 | 0.259 | 0.431 | 0.592 | 0.168 | 67 |
| ATOM | 0.661 | 0.867 | 0.703 | 0.648 | 0.642 | 0.825 | 0.515 | 0.576 | 0.590 | 0.204 | 0.401 | 30 |
| SiamRPN++ | 0.696 | 0.914 | 0.733 | 0.694 | 0.613 | 0.807 | 0.496 | 0.569 | 0.600 | 0.234 | 0.414 | 35 |
| DiMP | 0.660 | 0.859 | 0.723 | 0.666 | 0.643 | 0.821 | **0.532** | **0.581** | 0.594 | **0.182** | 0.402 | 57 |
| Manually designed rule-based | 0.544 | 0.716 | 0.416 | 0.402 | 0.453 | 0.618 | 0.283 | 0.302 | 0.470 | 0.682 | 0.158 | 19 |
| DTNet (ACT+SiamFC) | 0.649 | 0.875 | 0.541 | 0.580 | 0.519 | 0.710 | 0.352 | 0.342 | 0.520 | 0.329 | 0.303 | 26 |
| DTNet (ACT+CFNet) | 0.655 | 0.880 | 0.537 | 0.582 | 0.519 | 0.709 | 0.356 | 0.315 | 0.521 | 0.389 | 0.283 | 24 |
| DTNet (FCT+CFNet) | 0.643 | 0.861 | 0.546 | 0.601 | 0.513 | 0.697 | 0.353 | 0.332 | 0.496 | 0.338 | 0.283 | 32 |
| DTNet (FCT+SiamFC) | 0.660 | 0.891 | 0.610 | 0.583 | 0.533 | 0.731 | 0.360 | 0.341 | 0.518 | 0.277 | 0.300 | 36 |
| DTNet (ACT+FCT+SiamFC) | 0.665 | 0.893 | 0.621 | 0.585 | 0.539 | 0.726 | 0.364 | 0.342 | 0.521 | 0.287 | 0.298 | 23 |
| DTNet (ATOM+SiamRPN++) | **0.701** | **0.916** | **0.737** | **0.698** | **0.649** | **0.831** | 0.516 | 0.579 | **0.604** | 0.197 | **0.418** | 27 |

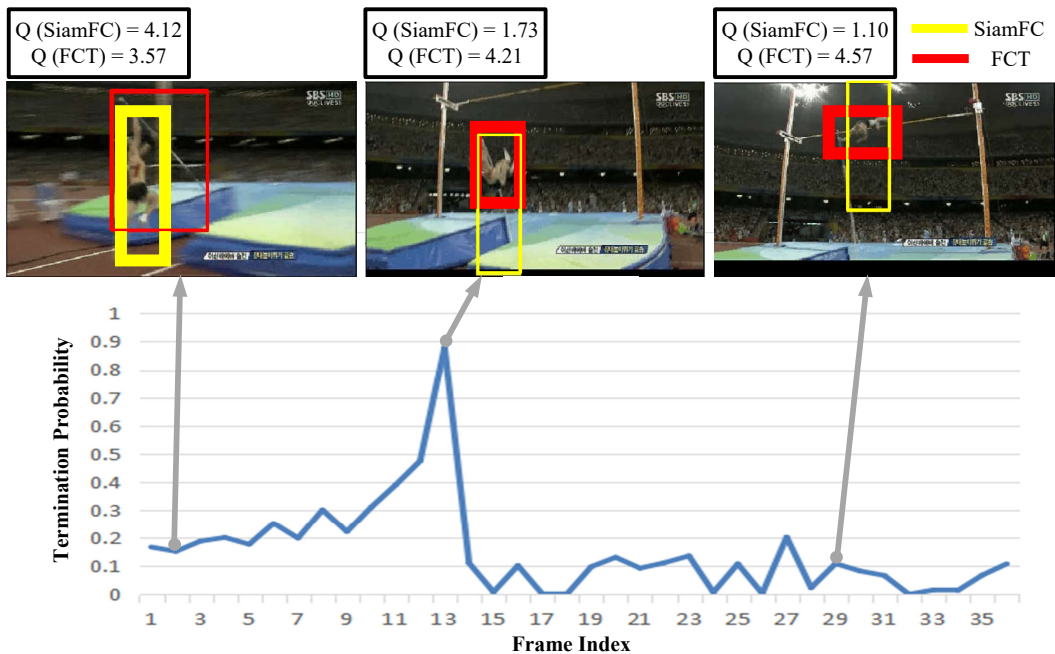

Figure 5: Visualization of the decision module using three sample frames within the training phrase.

it in the second frame according to the $Q$ value. Thus it leads to a high probability to terminate the current template tracker and switch to the detection tracker. And in the third frame, since the detection tracker outperforms the template tracker again, the termination probability remains low and thus the detection tracker is still in use as desired.

**Comparison with fusion-based trackers.** We further compare our DTNet with some trackers based on the fusion strategy. According to the quantitative results listed in Table 2, our method exhibits the best performance among real-time trackers on all four datasets. By associating Table 1 with Table 2, we find that although either FCT or SiamFC alone is outperformed by some state-of-the-art fusion-based trackers such as HSME (on OTB-2013) and MCCT-H (on OTB-2013 and OTB-100), the DTNet that combines them in a switching manner through the decision module performs significantly better than them. Such a finding demonstrates that the switching-based combination delivered by the decision module of the proposed DTNet is superior to the fusion-based combination that is broadly adopted by the existing state-of-the-art trackers.

Table 2: Comparison with the fusion-based trackers in terms of AUC

| Trackers / Benchmarks | Staple [3] | BranchOut [14] | HSME [20] | MCCT-H [35] | MEEM [40] | DTNet |
|---|---|---|---|---|---|---|
| OTB-2013 | 0.600 | - | 0.671 | 0.664 | 0.566 | **0.681** |
| OTB-50 | - | - | - | - | 0.473 | **0.626** |
| OTB-100 | 0.578 | **0.678** | 0.627 | 0.642 | 0.530 | 0.660 |
| LaSOT | 0.243 | - | - | - | 0.257 | **0.360** |

## 5 Conclusions

In this paper, we proposed an ensemble framework, namely DTNet, composed of a decision module and a tracker module for visual tracking. By HRL, the decision module enables the detection tracker and the template trackers that form the tracker module to compete with each other so that the DTNet can switch between them for different scenes. Differing from the fusion-based methods, the DTNet could learn an online decision to pick a particular tracker for a particular scene. Besides, we presented a new proposal-free detection tracker, which does not require the proposal of candidate bounding boxes of the target and thus makes the discriminating course flexible. Extensive results on several benchmarks demonstrated the superiority of the proposed DTNet over existing methods.

## Broader Impact

In this paper, the authors introduce DTNet which learns an online decision for switching to a proper tracker to conduct visual tracking in the current video frame. Although this paper only validates the efficacy of the decision learning framework in the specific scenario of visual tracking, it can actually be extended to other video-based computer vision tasks such as person re-identification, motion caption and action recognition, etc. It can be applied by defining a reward concerning the specific task and replacing the two trackers used in this paper with some other algorithms. To this end, the proposed DTNet could be of broad interest in different fields such as transportation industry, film industry, sport industry, etc.

As a method for visual tracking, the DTNet can inevitably be used for monitoring and security purpose. As a learning-based method, what the DTNet can track, a person or a pet, essentially depends on the training data. Therefore, the risk of applying our method to some tasks that could raise ethical issues can be mitigated by imposing a strict and secure data protection regulation such as the GDPR. Without a sufficiently large amount of data of high quality that contain the particular target, the DTNet cannot deliver a good tracking in the particular task.

## Acknowledgements

We acknowledge the support of the National Key Research and Development Plan of China under Grant 2017YFB1300205, the National Natural Science Foundation of China under Grants 61991411 and U1913204, the Shandong Major Scientific and Technological Innovation Project 2018CXGC1503, the Young Taishan Scholars Program of Shandong Province No.tsqn201909029 and the Qilu Young Scholars Program of Shandong University No.31400082063101.

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
