[Supplementary Material · NIPS2020_supp.pdf]

# Supplementary Materials of "Online Decision Based Visual Tracking via Reinforcement Learning"

## 1 More algorithm details

**Training details.** Algorithm 1 gives the details of the training process of the DTNet.The initial target is sampled around the ground truth randomly in the first frame. In each frame, the operation of flipping, random scaling and decoloring are conducted for data augmentation. The parameters of the decision and the tracker modules are all initialized through normal distribution.

The core component of Algorithm 1 is between line 16 and line 31. It can be seen that the training is divided into two phrases at line 16. In the first $n_\kappa$ episodes, both the decision and the tracker modules are updated while after that the latter is fixed with the former still updated. It is worth noting that for every $\tau$ steps, we clone the switch network $Q_\Omega$ to obtain a target network $\widehat{Q}_\Omega$ and use $\widehat{Q}_\Omega$ for generating the Q-learning targets $y_i$ for the succeeding $\tau$ steps of $Q_\Omega$. In each frame, the sample pair $\{I_t, G_t\}$ is collected in the sample replay $C_{\omega^*}$, where $\omega^*$ is the unselected tracker in the current frame. The samples in $C_{\pi i}$ are used to train the trackers $\pi_i (i = 1, 2)$. In other words, we keep on training the worse tracker in a scene to explore the scenes that both trackers are good at. After $n_\kappa$ episodes, either of the two trackers has gained a relatively reliable capability of tracking the target in a scene that it is suitable with. In this phrase, the most influential factor of the tracking performance lies in the selection of the tracker rather than the small update of the trackers. Thus we stop updating the tracker module to focus on the learning of the decision module. As an appealing byproduct, such a scheme significantly accelerates the training. In other words, freezing the parameters of the tracker module could enable the replay memory to train the decision module in an off-policy manner. The update also follows lines 17-21 of Algorithm 1 while the difference is that the samples are a batch with size 128 chosen from the replay memory while in the first $n_\kappa$ episodes the batch size is set to 1 .

**Decision module setup.** Fig. 1 shows the architecture of the switch and the termination networks that constitute the decision module. They are both composed of 3 convolutional layers containing 32 filters with size 8×8 and stride of 4, 64 filters with size 4×4 and stride of 2, and 64 filters with size 3×3 and stride of 1, respectively. Then, through a 512-dimensional fully-connected layer, the switch and the termination networks output the Q-value determining which tracker should be switched to and the termination probability of the current tracker respectively.

## 2 More visualization results

Fig. 2 shows extra visualization results of the DTNet in a style similar to Fig. 5 of the main submission. The visualizations further validate the perspective that different trackers are good at tracking the targets in different scenes: the template tracker such as SiamFC successfully tracks the targets which suffer from occlusion (rows 1 and 4) but fails in the scenes that contain targets subject to shape deformation (row 2) or change of view (row 3); on the other hand, the detection trackers such as FCT exhibits a contrary behavior. These visualizations justify the motivation of the decision module which essentially carries out a stitching strategy. They also demonstrate that such a strategy delivered through the switch and the termination networks works well for tracking various targets. We also show extra visualization results of various scenes in the **supplementary video**.

---

**Algorithm 1:** Training of the DTNet

---

**Input:** The video sequences for training and the corresponding ground truth, i.e. the bounding box of the target in the first frame of a video sequence

**Output:** The learned weights of the DTNet

**1** Initialize the switch network $Q_\Omega$, the termination network $\beta$ with parameters $\theta$, $\nu$;

**2** Initialize the target network $\widehat{Q}_\Omega$ with parameters $\widehat{\theta} = \theta$;

**3** Initialize the replay memory $D$ to capacity $N$

**4** Initialize the trackers $\pi_i$ and the training samples replay $C_{\pi i} \leftarrow \{\}(i = 1, 2)$ ;

**5** **for** $episode = 1, 2, .., n_\kappa, n_{\kappa+1}, ..n$ **do**

**6**      Randomly select a sequence $I = \{I_1, I_2, .., I_T\}$ with ground truth $G = \{G_1, G_2, .., G_T\}$;

**7**      $\Phi^* \leftarrow \varphi(I_1)$ ;

**8**      **for** $t = 2,..,T$ **do**

**9**          $\Phi^t \leftarrow \varphi(I_t)$ ;

**10**          $s_t \leftarrow concatenate(\Phi^*, \Phi_t)$;

**11**          **if** $(t == 2)$ :

**12**             Switch to $\omega_t$ via $\epsilon$-greedy policy over $Q_\Omega(s_t)$

**13**          Run $\pi_\omega$, observe $s'_t, r_t$;

**14**          **if** $\beta_{\omega,\nu}$ *terminates in* $s'_t$:

**15**             Switch new option $\omega$ via $\epsilon$-soft$(Q_\Omega(s'_t))$;

**16**          **if** $episode \leq n_\kappa$ **then**

**17**             **Update the switch network:**

**18**             $\delta \leftarrow r_t + \gamma[(1 - \beta_{\omega,\nu}(s'_t))\widehat{Q}_\omega(s'_t, \omega) + \beta_{\omega,\nu}(s'_t)\widehat{V}_\Omega(s'_t)]$;

**19**             $Q_\Omega(s_t, \omega; \theta) \leftarrow Q_\Omega(s_t, \omega; \theta) + \alpha_\theta \delta$;

**20**             **Update the termination network:**

**21**             $\nu \leftarrow \nu - \alpha_\nu \frac{\partial \beta_{\omega,\nu}(s'_t)}{\partial \nu}(Q_\Omega(s'_t) - V_\Omega(s'_t)))$;

**22**             **Update tracker module:**

**23**             $C_{\omega^*} \leftarrow \{I_t, G_t\}$;

**24**             Train the trackers $\pi_i(i = 1, 2)$ by $C_i$;

**25**          **else**

**26**             **Fix the parameters of the trackers**

**27**             Store the transition $(s_t, \omega_t, r_t, s'_t)$ in $D$;

**28**             Sample a mini-batch of transitions $(s_t, \omega_t, r_t, s'_t)$ from $D$;

**29**             **Update the switch network**;

**30**             **Update the termination network**;

**31**          **end**

**32**      **end**

**33**      Every $\tau$ steps update $\widehat{\theta} = \theta$;

**34** **end**

Figure 1: Architecture of the switch and the termination networks

Figure 2: Visualization results of the DTNet

# 3 Visual comparisons with the state-of-the-art methods

In this section, we provide visual comparisons in Fig. 3 which qualitatively demonstrate that the proposed DTNet outperforms the state-of-the-art tracking methods including MDNet [4], ECO [2], Staple [1] and CNN-SVM [3]. More visual comparisons with the state-of-the-art methods are available in the **supplementary video**.

Figure 3: Visual comparisons with the state-of-the-art methods