[Reviews · NeurIPS 2020]

Review 1

Summary and Contributions: The paper proposes a reinforcement learning framework (DTNet) for visual tracking. Unlike existing methods that either use a single tracker or fuse the outcomes of multiple trackers for each frame, the proposed method is essentially a selection scheme that learns online decisions to select an appropriate tracker for each individual frame. According to the results, the decisions are made wisely and thus as expected, the two trackers work jointly in a complementary manner. The idea of the online decision module for selecting a frame-specific tracker delivered by RL is interesting and forms the main contribution of this work.

Strengths: The motivation of the work is well explained in the introduction. The decision module of the DTNet is conceptually novel as it brings in a way to synergistically utilise multiple trackers that potentially have complementary performance over various frames for visual tracking. The idea of implementing the decision module as a termination network followed by a switch network also makes sense and the effectiveness of such an implementation through RL is well demonstrated by the experiments. The paper also presents an improved detection tracker in the context of the selection scheme and successfully incorporates it into the RL framework. Given the details provided in the paper, I believe the paper is technically sound as well. In addition, it is found that the DTNet seems efficient enough to deliver visual tracking at real-time rate according to the testing results. Therefore, this work could be of broad interest in the visual tracking community.

Weaknesses: Although the paper is technically sound, I have two issues about the design of the DTNet and would like to see the response from the authors. To deliver the decision module, why not directly use the switch network to make a decision on the selection of trackers? What is the point of introducing the termination network? Is it redundant or really useful? This work only explores how to select a tracker for each frame out of two trackers. I am curious if this method can be extended to select a tracker from more candidate trackers. For example, if three trackers are included, can the method still work with further improved performance?

Correctness: I am convinced that the proposed method including its algorithmic design and empirical methodology is technically correct as mentioned above. Most important claims such as “Either the detection or the template tracker is merely suitable for a particular group of scenes” in line 29 and “directly fusing the two types of trackers together is not wise” in line 37 are both theoretically discussed and experimentally validated. However, I also feel that there do exist two issues that might be less important to the overall quality of the paper but probably worth a further clarification in the authors response: First, in MCCT, each tracker is always affected by the tracking results of previous frames when tracking the object in the current frame. Thus for the proposed DTNet, is the tracking result of a frame produced through a particular tracker also affected by that of the previous frame produced by the same tracker? Second, if my understanding is correct, the termination module is designed to decide if the DTNet should keep using or terminate the current tracker, subject to a probability which indicates the tracking performance of the current tracker. But it seems that such a design could still lead to the termination of the current tracker even if it performs well on the current state. It would be interesting to know how often this happens and why it does not incur a problem in general.

Clarity: The difference between the proposed work and previous contributions is discussed in the introduction. I agree that most previous methods focus on how to fuse multiple trackers together to boost the performance while this paper advocates a selection strategy to make different trackers work alternatively.

Relation to Prior Work: The paper is well written and easy to follow.

Reproducibility: Yes

Additional Feedback: ------------------------Post Rebuttal---------------------- Overall, online decision by RL tracker selection is fine. The additional results provided have shown the effectiveness of switching to other SOTA trackers. This reviewer will keep the original rating.


Review 2

Summary and Contributions: This paper addresses the visual tracking task, in the generic setting. The authors introduce a technique to fuse a detection based and fusion based tracker. This is performed by learning a decision module, using reinforcement learning. It decides which tracker to use through a switch and a termination network. The authors also introduce a fully convolutional detection based tracker. Experiments are performed on the OTB and LaSOT datasets.

Strengths: - The paper is well written in terms of language.

Weaknesses: 1) I do not think that the authors are able to motivate or demonstrate the usefulness of their contributions. The proposed tracker performs far from current state-of-the-art trackers, such as SiamRPN++ and DiMP. I do not believe the proposed component to be effective in such more advanced trackers, since they integrate the advantages of both template and detection based method in a more unified manner. In order to demonstrate the usefulness, the authors should attempt to improve modern SOTA trackers, which is not done here. 2) The experiments are lacking in several aspects. Only two datasets are used, OTB and LaSOT (both OTB-2013 and OTB-50 are subsets of OTB-100 and should therefore not be considered). OTB is small and considered obsolete since results are saturated. In addition to LaSOT, the authors should experiment with large-scale datasets, such as TrackingNet, UAV123, or GOT10k. Moreover, the authors only compare with outdated trackers. The claim of state-of-the-art is wrong. 3) No deeper analysis or ablation of the method itself is performed. Only the fusion of different trackers is performed. 4) The motivation and method description is not that clear. Design choices are not motivated for the most part. 5) I could not find significant novelty in the proposed detection tracker FCT. It essentially seems to correspond to a fully-convolutional version of MDNet.

Correctness: There are issues regarding the state-of-the-art claim and experiments (see weaknesses).

Clarity: Lacking clarity (see weaknesses).

Relation to Prior Work: - The authors do not cover reinforcement learning works in visual tracking. - More recent tracking works (such as SiamRPN++ and ATOM, both CVPR 2019) which integrate advantages of both "detection" and "template" methods are not addressed or compared with.

Reproducibility: No

Additional Feedback: --------------- Post Rebuttal ------------------ Comments on the rebuttal: - The authors provide results of fusing existing SOTA trackers with the proposed switching strategy. On all datasets, results are marginally better than selecting the best of the two trackers (0.001 - 0.007 AUC/EAO). This improvement is rather minor, and difficult to put into context since other fusion methods are not compared. There are naive baselines for doing this, for example simply averaging the two bounding boxes. But exactly this topic has attracted substantial research interest over the years: MCCT, LCT, MEEM, [N. Wang, ICCV 2015], [B. Han, CVPR 2017], just to name a few. It is unclear to me if these results are competitive compared to such previous approaches, or even to naive baselines. - I consider my concern about comparisons on more datasets and to current SOTA trackers well addressed by the rebuttal. - I am not sure which version of the tracker we should compare the "Manually designed rule-based" to. - The novelty of the FCT is not one of my main concerns. The authors say that it is not the main contribution, and could change some formulations in the paper to address this. I have to following remaining concerns: A) My initial issue 3), about analysis and ablations, is unaddressed. I think this is a major weakness of the paper. Very little insight into the method is provided. B) As discussed in issue 4) of my initial review, I think the paper lacks clarity and the authors do not motivate design choices. C) I still do not believe that the authors manage to demonstrate the usefulness of their contributions (original issue 1). Although it is in the rebuttal shown to improve performance when fusing two SOTA trackers, baselines or previous works are not compared with. Moreover, these marginal improvements come as the cost of effectively doubling the computational complexity. D) I am also concerned that the integration "DTNet (ATOM+SiamRPN++)" would require major changes in the story and motivation of the paper, since in is case, the paper should be framed more as a general tracker fusion approach. Summary: As the rebuttal addresses some of my issues, mainly point 2), I am willing to raise my rating. But considering the other issues above, I still believe that the paper is below acceptance threshold.


Review 3

Summary and Contributions: This paper proposes an online decision based visual tracking framework and a proposal-free detection tracker. The organization and writing of this paper are unsatisfactory. The proposed method is adhoc and lacks novelty. The methods compared in the experiment section are slightly out-of-date.

Strengths: This major contribution of this paper is an online decision based visual tracking framework with reinforcement learning. Its key idea is to adaptively ensemble detection tracker and template tracker. The proposed method is intuitive and technically correct. The topic in this paper is vital for the real-world machine learning tasks, but the proposed method is slightly adhoc and lacks novelty.

Weaknesses: 1. In my opinion, "using different kinds of trackers to pursue more reliable tracking is not wise as they follow different tracking principles" is the cornerstone of this paper. However, the authors fail to demonstrate why this is not wise. In my opinion, combining different kinds of clues is expected to make the tracker stronger, especially when these clues are based on different principles. 2. Does the proposed online decision mechanism outperform a manually designed rule-based decision module? More comparisons are needed and essential for demonstrating the effectiveness of the proposed method. 3. Several claims are not well supported by experiments or enough demonstrations, such as the claim in lines 37-39. 4. The comparisons in this paper are not entirely fair. The previous methods in the experiment section are out-of-data. 5. Although the paper tackles the online decision framework and proposal-free detection problems simultaneously, both of the proposed methods lack novelty, making the entire ad hoc.

Correctness: Yes.

Clarity: Slightly unsatisfied.

Relation to Prior Work: No. The major claim of this paper is not convincing enough. The effectiveness of the proposed method is not demonstrated thoroughly, so the contributions of this paper might be insufficient.

Reproducibility: Yes

Additional Feedback:


Review 4

Summary and Contributions: The primary contribution of this paper is the ensemble framework which can switch between a template tracker and a detection tracker based on the tracking accuracy. This is driven by the observation that the two types of trackers are good at tracking the targets in different cases. Besides, a detection tracker without the need ofgenerating proposals is presented. Experimental results show that the proposed method improved the tracking performance by making good use of the advantages of both the detection and the template trackers.

Strengths: Overall, I am positive to the paper due to the following aspects of the work. 1) Prior methods (fusion-based methods) involving multiple trackers typically try to combine information extracted at feature level while the proposed method seeks information gathering at tracker level via a competing mechanism. The experimental results show that the proposed scheme of combination at tracker level outperforms the combination at feature level. 2) The proposed framework represents a novel ensemble strategy which essentially carries out a “predict-and-evaluate” tactic through Hierarchical Reinforcement Learning (HRL). The visualization of the decision module show this design reasonable and effective. 3) The authors build a fully convolutional tracker based on object detection which does not rely on the step of proposal generation, leading to an accurate and efficient detection of the bounding box of the target.

Weaknesses: The following concerns should be addressed to improve the paper. 1) The authors have validated the effectiveness of the proposed framework on OTB-50, OTB-100, OTB-2015 and LASOT datasets. Such an evaluation is good while all of these benchmarks evaluate the competing methods by the metrics of AUC and precision. So I suggest the authors perform additional evaluation on the VOT dataset to compare the methods in terms of different metrics like Accuracy, Robustness, and EAO. 2) It is nice to see the visualization of the decision module provided in Fig.5. I am also interested in some informative visual plot with regard to the cases listed in Eq. 7. In particular, how often the reward estimated via Eq. 7 during the switching process falls into the third case (i.e. neither tracker is able to track the target)?

Correctness: Generally, the claims related to the technical details made in the paper seem correct. I am only confused by Eqs. 8 and 9 which update the critic and the actor with the outputs from each other. This is inconsistent with the common ‘Actor-Critic’ algorithm where the critic uses Q-learning updates to estimate the action-value function and the actor updates parameters in the direction of the critic’s action-value gradient. Will this update scheme hinder the training?

Clarity: The presentation is generally good. Several symbols in the equations are not properly used and thus should be corrected. 1) It seems that R_t(S,omega) in Eq. 7 corresponds to r(s, omega) in Eq.1 which both represent the reward. Thus it’s better to make them consistent. 2) In line 173, the regression value r*/r ought to be replaced with other symbols to avoid the repetitive use of symbols. 3) Fig.4 is of low resolution and should be improved.

Relation to Prior Work: As mentioned above, this paper is different from prior fusion-based trackers. Besides, it reveals that RL can work well with time-dependent tracking tasks. The relevant papers have also been cited and discussed properly.

Reproducibility: Yes

Additional Feedback: Based on the authors' response and other reviewers' comments, I retain my rate.

[Author Response · NeurIPS 2020]

We thank reviewers for the valuable comments. Please see below for our responses to specific comments.

**R1: About the termination network.** Only using a switch network is not reliable as switching to another tracker does
not necessarily improve the tracking. Thus we tend to keep using a certain tracker as much as possible. Our termination
network makes the decision module avoid oscillating between the two trackers especially when they have similar
accuracy. In the training phrase, the tracker that actually performs well could suffer from an improper termination due
to the probability indicating its tracking performance. However, our termination scheme enforces the agent to explore
more states that would not have been selected. We also observe that the probability is typically close to either 0 or 1 at
the later stage of training, which means that improper termination hardly occurs at that stage.

**R1: Extend to more trackers?** Yes, our framework can be easily extended to more trackers. For instance, the results
of using 3 trackers including ACT, FCT and SiamFC are shown in the second row of the table below. Considering both
accuracy and efficiency, we use two trackers in the proposed DTNet.

| Method | OTB2015 | | TrackingNet | | UAV-123 | | LaSOT | | VOT18 | | | Speed (fps) |
|---|---|---|---|---|---|---|---|---|---|---|---|---|
| | AUC | Prec. | AUC | Prec. | AUC | Prec. | AUC | Prec. | Accuracy | Robustness | EAO | |
| DTNet (FCT+SiamFC) | 0.660 | 0.891 | 0.610 | 0.583 | 0.533 | 0.731 | 0.360 | 0.341 | 0.518 | 0.277 | 0.300 | 36 |
| DTNet (ACT+FCT+SiamFC) | 0.665 | 0.893 | 0.621 | 0.585 | 0.539 | 0.726 | 0.364 | 0.342 | 0.521 | 0.287 | 0.298 | 23 |
| DTNet (ATOM+SiamRPN++) | 0.701 | 0.916 | 0.737 | 0.698 | 0.649 | 0.831 | 0.516 | 0.579 | 0.604 | 0.197 | 0.418 | 27 |
| SiamRPN++ | 0.696 | 0.914 | 0.733 | 0.694 | 0.613 | 0.807 | 0.496 | 0.569 | 0.600 | 0.234 | 0.414 | 35 |
| ATOM | 0.661 | 0.867 | 0.703 | 0.648 | 0.642 | 0.825 | 0.515 | 0.576 | 0.590 | 0.204 | 0.401 | 30 |
| DiMP | 0.660 | 0.859 | 0.723 | 0.666 | 0.643 | 0.821 | 0.532 | 0.581 | 0.594 | 0.182 | 0.402 | 57 |
| Manually designed rule-based | 0.544 | 0.716 | 0.416 | 0.402 | 0.453 | 0.618 | 0.283 | 0.302 | 0.470 | 0.682 | 0.158 | 19 |

11
**R1: Tracking affected by previous frames?** No, the tracking delivered by the DTNet is not affected by the tracking
results of previous frames. The benefit is that the tracker is not influenced by inaccurate tracking on previous frames.

**R2: Comparison with latest SOTA trackers.** As suggested, we have compared our method with the latest SOTA
trackers such as SiamRPN++, ATOM and DiMP. Specifically, we replace the baseline trackers FCT and SiamFC with
SiamRPN++ and ATOM, and perform our decision module to make them compete with each other. The table above
shows that our DTNet still improves both SiamRPN++ and ATOM in all datasets. This is because in our method, two
baseline trackers could work alternatively to conduct tracking within different scenes that they are adept in. Such results
indicate that combining different trackers based on an intelligent switching scheme is superior over a single tracker
even if it is the SOTA which already integrates the advantages of both template and detection based trackers.

**R2: Evaluation on other datasets.** As shown in the table above, we have evaluated our DTNet with different baseline
trackers on other datasets suggested by the reviewers including OTB2015, TrackingNet, UAV-123, LaSOT and VOT18.
It can be seen that our method achieves consistent improvement over various datasets benefitting from the proposed
decision module which could select different types of trackers for handling different scenes.

**R3: Motivation.** We agree that combining different kinds of clues is expected to make the tracker stronger. In fact,
what we mean here is that such a fusion manner might not be the best choice. In this work, instead of fusing different
types of trackers into one, we advocate an intelligent switching strategy to make them coexist and compete with each
other for different scenes. To the best of our knowledge, this has never been explored before. The results in Table 2 of
the paper show that the proposed strategy can utilize the advantages of different types of trackers and produce significant
gains. Moreover, even with the two similar fusion-based trackers such as SiamRPN++ and ATOM, our method still
makes improvement as shown in the table above. This also shows the potential of our method in more general cases.

**R2: About the FCT tracker.** Please note that the FCT tracker is not the primary contribution of our work. Our
main contribution lies in the novel decision module which automatically selects a tracker to handle different scenes as
recognized by R1 and R4. Yes, FCT is extended from MDNet. Compared to MDNet, the proposed FCT uses pixel-level
classification and regression which does not require the expensive proposal generation. We prefer such a proposal-free
tracker as it does not affect the efficiency of the whole ensemble framework much.

**R3: Compare with manually designed rule-based decision module.** We have included the manually designed rule-
based decision module for comparison. It is implemented by picking a particular tracker based on the confidence score
of tracking subject to the thresholds set manually. The results are given in the bottom row of the table above. Apparently,
our automated decision module significantly outperforms such a handcrafted one which relies on handcrafted thresholds
for tracker selection. Besides, our method is more efficient as it only performs each tracker once in the decision-making
process while the handcrafted module has to carry out both trackers and use their output confidence scores for decision.

**R4: About update scheme of the decision module.** The actor of the update scheme defined by Eq. 9 outputs a
probability to indicate how possible the tracker should be terminated. The critic defined by Eq.8 is then updated subject
to this termination probability when evaluating the value of a state-option pair. And the actor still updates parameters in
proportion to the action-value gradient of the critic.

**R4: Inconsistent symbols in Eq. 7 and line 173.** Thx. We have made corrections to avoid using inconsistent symbols.

[Meta-Review · NeurIPS 2020]

The initial scores were 3478. the main concerns were: 1) should combine the framework with SOTA trackers; 2) insufficient experiments on larger datasets and current methods; 3) lack fo deeper ablation study; 4) comparison with hand-designed rules; 5) incremental novelty. In the response, authors provide experiment results fusing SOTA trackers on the larger datasets, compared with SOTA trackers, showing improved performance. Authors also provide ablation study using hand-designed rules. During discussion, R2 was satisfied with the new comparisons with SOTA and larger datasets, but was not convinced that the fusion method was useful since there was no ablation study comparing only fusion methods (while keeping trackers the same). R3 was mostly satisfied with the response, but novelty concern was not addressed fully. After the discussion, R2 and R3 upgraded to 4 and 6, leaving the final score as 4678. 3 out fo 4 reviewers are positive on the paper, while R2 was mainly concerned about the ablation study comparing only fusion methods, while keeping baseline trackers the same. To be fair to the authors, the AC notes that in the original review, R2 did not state specifically what type of deeper ablation study was needed, thus making it difficult for the authors to address. Nonetheless, the method is compelling and improves SOTA. Thus, the AC recommends accept. Authors should update the paper according to the reviews/responses, including the missing ablation study